# A New Gold(III) Complex, TGS 703, Shows Potent Anti-Inflammatory Activity in Colitis via the Enzymatic and Non-Enzymatic Antioxidant System—An In Vitro, In Silico, and In Vivo Study

**DOI:** 10.3390/ijms24087025

**Published:** 2023-04-10

**Authors:** Jakub Włodarczyk, Julia Krajewska, Łukasz Szeleszczuk, Patrycja Szałwińska, Agata Gurba, Szymon Lipiec, Przemysław Taciak, Remigiusz Szczepaniak, Izabela Mlynarczuk-Bialy, Jakub Fichna

**Affiliations:** 1Department of Biochemistry, Faculty of Medicine, Medical University of Lodz, Mazowiecka 5, 92-215 Lodz, Poland; jakub.wlodarczyk@stud.umed.lodz.pl (J.W.); julia.krajewska@stud.umed.lodz.pl (J.K.); patrycja.szalwinska@stud.umed.lodz.pl (P.S.); 2Department of General and Oncological Surgery, Medical University of Lodz, Pomorska 251, 92-213 Lodz, Poland; 3Department of Organic and Physical Chemistry, Faculty of Pharmacy, Medical University of Warsaw, Banacha 1, 02-093 Warsaw, Poland; lukasz.szeleszczuk@wum.edu.pl; 4Department of Pharmacodynamics, Faculty of Pharmacy, Medical University of Warsaw, Banacha 1 Str., 02-093 Warsaw, Poland; agata.grabowska@wum.edu.pl (A.G.); przemyslaw.taciak@wum.edu.pl (P.T.); 5Department for Histology and Embryology, Medical University of Warsaw, Chalubinskiego 5, 02-004 Warsaw, Poland; szymon.lipiec@student.wum.edu.pl (S.L.); izabela.mlynarczuk-bialy@wum.edu.pl (I.M.-B.); 6Inwex Ltd., Solidarnosci 34, 25-323 Kielce, Poland; remigiusz.szczepaniak@inwex.pl

**Keywords:** inflammatory bowel disease, gold, anti-inflammatory

## Abstract

Inflammatory bowel diseases (IBD) and their main representatives, Crohn’s disease and ulcerative colitis, are worldwide health-care problems with constantly increasing frequency and still not fully understood pathogenesis. IBD treatment involves drugs such as corticosteroids, derivatives of 5-aminosalicylic acid, thiopurines, and others, with the goal to achieve and maintain remission of the disease. Nowadays, as our knowledge about IBD is continually growing, more specific and effective therapies at the molecular level are wanted. In our study, we tested novel gold complexes and their potential effect on inflammation and IBD in vitro, in silico, and in vivo. A series of new gold(III) complexes (TGS 404, 512, 701, 702, and 703) were designed and screened in the in vitro inflammation studies. In silico modeling was used to study the gold complexes’ structure vs. their activity and stability. Dextran sulphate sodium (DSS)-induced mouse model of colitis was employed to characterize the anti-inflammatory activity in vivo. Lipopolysaccharide (LPS)-stimulated RAW264.7 cell experiments proved the anti-inflammatory potential of all tested complexes. Selected on the bases of in vitro and in silico analyses, TGS 703 significantly alleviated inflammation in the DSS-induced mouse model of colitis, which was confirmed by a statistically significant decrease in the macro- and microscopic score of inflammation. The mechanism of action of TGS 703 was linked to the enzymatic and non-enzymatic antioxidant systems. TGS 703 and other gold(III) complexes present anti-inflammatory potential and may be applied therapeutically in the treatment of IBD.

## 1. Introduction

Gold was used for medical purposes for thousands of years with the first mentions in ancient Chinese and Arabic medicines [1]. Nowadays, gold gets a growing interest among scientists, especially in a form of nanoparticles (AuNP) and gold(I) and (III) complexes.

Gold nanoparticles, first presented by Faraday in 1857, are believed to have versatile properties and can be used in clinical chemistry, bioimaging, cancer therapy, and drug delivery as they can be specifically designed to possess a particular profile [2]. In anticancer interventions, AuNPs can be used in photothermal therapy (PTT, intervention with the use of electromagnetic radiation to generate heat for destruction of cancer cells) or radiofrequency therapy (RFA, radiofrequency ablation of cancer tissue by heat with the use of medium frequency alternating current). Another use of AuNPs is as drug carriers and, potentially, modulators of angiogenesis.

The era of metallodrug complexes began in the 1960s with the discovery of the anticancer properties of cisplatin [3]. Platinum-based drugs are widely used in the treatment of various types of cancer, particularly head and neck, testicular, and ovarian cancers. However, due to many severe side effects, the search for an alternative to platinum began, and similarities between platinum(II) and gold(III) were noticed: gold(III) has a 5d^8^ electronic configuration and is thus isoelectronic with platinum(II) and, likewise, forms four-coordinated square-planar complexes. This led to the belief that gold(III) complexes may have anticancer properties similar to platinum metallodrugs. The use of chelating ligands with nitrogen donors and cyclometalated structures which prevent reducing gold oxidation state from (III) to (I) allowed its application for chemotherapeutic purposes.

Lately, increasing attention has been paid to gold complexes and particles as potential anti-inflammatory therapeutic options. Their role has been discussed in terms of retinopathy, Alzheimer’s disease, skin disorders, and vascular and metabolic diseases [4]. Moreover, gold was proven to alleviate the inflammation linked with the gastrointestinal system. In an in vitro study, Au nanoparticles reversed the epithelial barrier dysfunction and inflammatory response in colonic epithelial cells, NCM460, after stimulation with lipopolysaccharide (LPS) [5]. The authors of that study suggested that a possible way of action was through inhibition of NF-κB and the ERK/JNK pathway, which are crucial in inflammation. Fujita et al. evaluated the effect of gold nanoparticles on the inflammatory profile of murine macrophages stimulated with LPS [6]. Authors demonstrated that gold nanoparticles were able to significantly decrease the levels of pro-inflammatory cytokines (TNF-a and IL-1β).

Recently, we evaluated a novel gold(III) complex—TGS 121 ([Au(CN)_4_]_2_ClO_2_^−^)—as an anti-inflammatory therapeutic [7,8]. In this study, we designed a series of novel gold(III) complexes: TGS 404, 512, 701, 702, and 703. We examined their anti-inflammatory potential in vitro, i.e., in the lipopolysaccharide (LPS)-stimulated RAW 264.7 macrophages, and in vivo, i.e., in the dextran sulphate sodium (DSS)-induced mouse model of colitis. These studies were supported by in silico and ex vivo analyses to elucidate the mechanism of action of novel complexes.

To explain, at the molecular level, both the mechanism of action as well as the differences in biological effects resulting from the excipients present in the particular composition, molecular modeling methods are frequently used. There are two main groups of molecular modeling methods, they are based on either molecular mechanics (MM) or quantum mechanics (QM). While the MM methods are significantly less computationally demanding, this is at the cost of their accuracy, in comparison to the QM methods. This is particularly important when modeling the metal–organic systems, such as the one in this study, for which the MM are usually not very well parametrized. Therefore, in the current work, we have decided to use the QM methods to increase the accuracy and credibility of the obtained results.

## 2. Results

### 2.1. TGS Complexes Showed an Anti-Inflammatory Profile In Vitro in RAW264.7 Macrophages

The in vitro studies consisted of a neutral red uptake (NRU) test to verify the cytotoxicity of the compound and a Griess test to assess the NO production.

In the case of TGS 404 and TGS 512, the NRU test showed no decrease in cell viability up to the concentration of 5 × 10^−7^ M (Figure 1A), while in TGS 701, 702, and 703 the decrease in cell viability was observed at 1 × 10^−3^ M concentration (Figure 1B). As the next step, we assessed how the NO production in RAW264.7 was influenced by the treatment with TGS compounds. In the Griess test, NO production is evaluated based on the concentration of nitrite in cell culture supernatants. The anti-inflammatory efficacy of the studied compounds is assessed at concentrations that do not cause a decrease in cell viability. We observed that NO production was significantly decreased by TGS 404 at the concentration 1 × 10^−7^ M (down to 76.88 ± 6.64%, where 100% is the value for untreated, LPS-stimulated cells) and TGS 512 at the concentration 5 × 10^−8^ M (down to 81.13 ± 4.33). TGS 701, 702, and 703 significantly decreased the NO production at the concentration 1 × 10^−4^ M (down to 28.68 ± 1.93; 50.67 ± 5.19; and 27.78 ± 9.88, respectively). For each compound, the inhibition of NO production was concentration-dependent (Figure 1C,D).

### 2.2. Molecular Modeling Suggests Higher Pharmacological Activity and Stability of Complexes TGS 701, 702, and 703 over TGS 404 and 512

TGS 512

Cyanide (CN^−^) is a highly basic and small ligand, hence it readily saturates the coordination sphere of metal ions. The resulting cyanometallate anions, Me(CN)_m_^n−^, are often used as ligands for building more complex structures. Homoleptic cyanometallates refer to complexes where the only ligand is cyanide. For the first-row transition metals, well-known homoleptic cyanometallates are the hexacyanides. Several tetracyanometalates are also known, the best known being those of the d8 metals, Ni(II), Pd(II), and Pt(II). These species are square-planar and diamagnetic [9].

The transition metal can also form isocyanide complexes, Me(NC)_m_^n−^, in which the metal is bonded directly to nitrogen instead of carbon. For example, Rayón et al. have shown that the cyanide arrangement is preferable for late transition metals (Co–Zn) and Cr, whereas early transition metals (Sc, Ti, and V), as well as Mn and Fe, show a preference for the isocyanide isomer [10]. Nevertheless, it must be pointed out that in the cases of V, Mn, and Fe both isomers lie very close in energy. Therefore, transition metal cyanides exhibit linkage isomerism, defined as the existence of coordination compounds that have the same composition differing with the connectivity of the metal to a ligand, in this case, either via C or N.

Our aim was not only to make predictions for the structure of TGS 512 (Na[Au(CN)_4_]) complex but also to study the competition between the cyanide and isocyanide arrangement as a function of the number of ligands both in vacuo and in different solvent media. We have searched for stable structures corresponding to the homoleptic cyanide complex [Au(CN)_4_]^−^, the homoleptic isocyanide compound [Au(NC)_4_]^−^, as well as the heteroleptic complexes [Au(CN)_x_(NC)_y_]^−^, x + y = 4. It should be noted here that for the [Au(CN)_2_(NC)_2_]^−^ two isomers are possible. The first one is symmetric (S), in which the CN^−^ ligands are opposite to each other, and the second is non-symmetric (NS), in which the CN^−^ groups are next to each other. This resulted in six possible linkage isomers, which were geometry optimized both in vacuo as well as through the use of the implicit solvent method for water and DMSO solutions. The results of the calculations are presented in Table 1 and in Figure 2, Figure 3, Figure 4 and Figure 5. The most stable isomer, Au(CN)_4_^−^, is presented in Figure 2.

All of the optimized isomers were found to be square planar, with the clear energetic and thermodynamic preference of the homoleptic cyanide complex [Au(CN)4]^−^ in all three studied environments. Substitution of one cyanide ligand by isocyanide was determined as endothermic, with required energy of c.a. 17 kcal/mol per one isocyanide ligand, increasing with the number of already substituted CN^−^ groups. Among the studied isomers, the Au–N distances are generally shorter than Au–C distances, and the C–N distances are generally shorter for the cyanide isomers than for isocyanide ones. 

To investigate the energy changes resulting from complexation, calculations were performed according to the equation ∆A = A(Au(CN)4^−^) − [A(Au(III)) + 4A(CN^−^)], where “A” is either E, E + ZPVE, or ΔG. The results of those calculations are presented in Table 1.

The formation of Au(CN)4^−^ is exothermic and spontaneous, due to the negative values of both energy and Gibbs free energy changes during the complex formation.

TGS 404

Diorganosulfoxides, such as dimethyl sulfoxide (DMSO), are frequently encountered ligands in transition metal complexes [11]. For the majority of transition metals, the coordination occurs via the hard oxygen atom, whereas in a smaller number of complexes, the softer sulfur atom is involved in the coordination of the transition metal [12]. In general, the harder 3d-transition metals prefer coordination by oxygen, whereas the softer 4d- and 5d-transition metals have a higher affinity to sulfur [13]. Therefore, the complexes involving DMSO as a ligand can exhibit linkage isomerism, similar to complexes including cyanide.

Our aim was not only to make predictions for the structure of [Au(CN)_4_]^−^ complex with DMSO but also to study the competition between the cyanide and DMSO as ligands both in vacuo and in DMSO as solvent.

First, we searched for stable structures corresponding to the homoleptic cyanide-DMSO complex of Au(III): [Au(CN)_3_DMSO]. We assumed that in that complex only cyanide ligands will be present, and not isocyanide, due to the stability of the cyanide complex (Table 2). It should be noted here that for the [Au(CN)_3_DMSO] two isomers are possible. In the first one, the coordination occurs via the sulfur atom (κ-S) and in the second via oxygen (κ-O).

To investigate the energy changes resulting from the substitution of cyanide by DMSO, calculations were performed according to the equation ∆A = A([Au(CN)_3_DMSO]) +A(CN^−^) − [A(Au(CN)_4_^−^ + A(DMSO)], where “A” is either E, E + ZPVE, or ΔG. The results of those calculations are presented in Table 3, and the most stable isomer, [Au(CN)_3_DMSO] κ-O, is presented in Figure 3.

The formation of every possible [Au(CN)_3_DMSO] isomer is endothermic and nonspontaneous, due to the positive values of energy and Gibbs free energy changes during the complex formation. For such complexes, the κ-O isomers are preferable, which is especially visible for the in vacuo models. The highly positive ΔG of formation for κ-O complex in vacuo results from the decrease of entropy, as the isomer dynamics of methyl groups are significantly hampered by the nearby presence of cyanide ions. Lower values have been obtained using the PCM solvation model, which indicates that such complexes are more stable in the DMSO; however, even in such cases, the substitution of cyanide by the DMSO in the Au(CN)_4_^−^ is thermodynamically not preferable. This proves that Au(CN)_4_^−^ is a stable complex even in DMSO.

However, very stable Au(CN)_4_^−^ might form complexes with DMSO by increasing the coordination number of Au(III) to five or even six, which would lead to the formation of [Au(CN)_4_DMSO]^−^ and [Au(CN)_4_DMSO_2_]^−^ complexes, respectively. To verify that hypothesis, we have searched for stable structures corresponding to the homoleptic cyanide-DMSO complex of Au(III): TGS 404 ([Au(CN)_4_DMSO]^−^) and [Au(CN)_4_DMSO_2_]^−^. We have assumed that in those complexes only cyanide ligands will be present, and not isocyanide, due to the stability of the tetracyanide complex (Table 2). It should be noted here that for the [Au(CN)_4_DMSO_2_]^−^ three isomers are possible. In the first one, the coordination occurs via sulfur atom (κ-S) with both DMSO molecules, in the second via oxygen (κ-O) with both DMSO molecules, and in the third complex one DMSO molecule is κ-O and the second is κ-S. For the [Au(CN)_4_DMSO]^−^ two isomers are possible. In the first one, the coordination occurs via the sulfur atom (κ-S) and in the second via oxygen (κ-O).

To investigate the energy changes resulting from the complex formation with one DMSO ligand, calculations were performed according to the equation ∆A = A([Au(CN)_4_DMSO]^−^) − [A(Au(CN)_4_^−^ + A(DMSO)], where “A” is either E, E + ZPVE, or ΔG. The results of those calculations are presented in Table 4, and the most stable isomer, [Au(CN)_4_DMSO]^−^ κ-S, is presented in Figure 4.

Negative values of E and E + ZPVE indicate that intermolecular interactions between Au(CN)_4_^−^ and DMSO can stabilize that kind of complex. The implicit solvent DMSO model results suggest that the κ-O and κ-S isomers are energetically similar; however, in vacuo calculations show the preference for κ-S isomer, which is opposite to the results obtained for Au(CN)_3_DMSO. Positive values of ΔG indicate that the complexes of Au(CN)_4_^−^ and DMSO are thermodynamically not stable. Despite the intermolecular interactions occurring between the Au(CN)_4_^−^ and DMSO, the decrease in entropy destabilizes those complexes. This entropic penalty results from the reduction of dynamics of DMSO when it becomes the ligand.

To investigate the structure and stability of the complex formed when two DMSO ligands react with A(Au(CN)_4_^−^, calculations were performed according to the equation ∆A = A([Au(CN)_4_DMSO_2_]) − [A(Au(CN)_4_^−^ + 2A(DMSO)], where “A” is either E, E + ZPVE, or ΔG. The results of those calculations are presented in Table 5, and the most stable isomer, [Au(CN)_4_DMSO_2_]^−^ κ-S, S, is presented in Figure 5.

Concluding, the substitution of CN^−^ by DMSO to form Au(CN)_3_DMSO is both energetically and thermodynamically not preferable due to the high stability of Au(CN)_4_^−^. When gold(III) tetracyanide is dissolved in DMSO, intermolecular forces between the Au(III) and sulfur atom (Lewis base) can be observed. However, those interactions are too weak to form thermodynamically stable complexes in which the coordination number of Au(III) would exceed four. 

TGS 701, TGS 702, and TGS 703

The next series of calculations were designed to understand and explain the structure and behavior of Au(CN)_4_^−^ in a water environment. As has been stated above (Table 2), gold(III) tetracyanide is stable in water solutions. Nevertheless, to check the possibility of dissociation of this complex, resulting in the formation of gold(III) aquo tricyanide ([Au(CN)_3_]H_2_O), calculations were performed according to the equation ∆A = [A([Au(CN)_3_]H_2_O) + A(CN^−^)] − [A(Au(CN)_4_^−^) + A(H_2_O)], where “A” is either E, E + ZPVE, or ΔG. The results of those calculations are presented in Figure 6 and Table 6.

The formation of [Au(CN)_3_H_2_O] is endothermic and nonspontaneous, due to the positive values of energy and Gibbs free energy changes during this complex formation. Lower values have been obtained using the PCM solvation model, which indicates that such complexes are more stable in water; however, even in such cases, the substitution of cyanide by the water molecules in the Au(CN)_4_^−^ is thermodynamically not preferable. This proves that Au(CN)_4_^−^ is a stable complex and does not dissociate in water.

However, very stable Au(CN)_4_^−^ might form complexes with water molecules by increasing the coordination number of Au(III) to five or even six, which would lead to the formation of [Au(CN)_4_H_2_O]^−^ and [Au(CN)_4_(H_2_O)_2_]^−^ complexes, respectively. To verify that hypothesis we have searched for stable structures corresponding to the homoleptic tetracyanide-(di)aquo complexes of Au(III): [Au(CN)_4_H_2_O]^−^ and [Au(CN)_4_(H_2_O)_2_]^−^. We have assumed that in those complexes only cyanide ligands will be present, and not isocyanide, due to the stability of the tetracyanide complex (Table 1). To investigate the energy changes resulting from the complex formation with water as a ligand, calculations were performed according to the equations ∆A = A([Au(CN)_4_H_2_O]^−^) − [A(Au(CN)_4_^−^ + A(H_2_O)], ∆A = A([Au(CN)_4_(H_2_O)_2_]^−^) − [A(Au(CN)_4_^−^ + 2A(H_2_O)], where “A” is either E, E + ZPVE, or ΔG. The results of those calculations are presented in Table 7 and Figure 7.

Negative values of E and E + ZPVE indicate that intermolecular interactions between Au(CN)_4_^−^ and water can stabilize that kind of aquo complex. The implicit solvent water model results suggest that diaquo complexes should be more energetically stable; however, in vacuo calculations show that the difference is irrelevant. Positive values of ΔG indicate that the complexes of Au(CN)_4_^−^ and water are thermodynamically not stable, especially diaquo complexes. Despite the intermolecular electrostatic interactions occurring between the positively charged Au atom of Au(CN)_4_^−^ and oxygen atoms of water, the decrease of entropy destabilizes those aquo complexes. This entropic penalty results from the reduction of dynamics of water when it becomes the ligand.

Partition coefficients reflect the solute hydrophobicity, partitioning between different solvents or pharmacokinetic characteristics. Undoubtedly, the most widely used partition coefficient is the octanol/water partition coefficient (hence referred to as logP_OW_). The amphiphilic nature of the octanol molecules, with a polar head group attached to a flexible nonpolar tail, affords them similar characteristics to the main constituents of biological membranes.

In addition, hydrophobicity is conventionally expressed by the logP_OW_ value, where a positive value of this ratio (for the case of a lipophilic substance) reflects a preference for the organic phase, while a negative value (for the case of a lipophobic substance) indicates an affinity for water. Hydrophobicity is a key descriptor used to assess and model drug partitioning and pharmacokinetic characteristics; it may be useful to estimate the solubility of a solute in a solvent or used as a measure of the bioconcentration factor.

The partition coefficients are tightly related to the electronic structure of the solvated molecules; therefore, the methods of quantum chemistry can be also employed in this type of research [14]. The theoretical logP_OW_ value for Au(CN)_4_^−^ was calculated according to the equation: logP_OW_ = (ΔG_water_ − ΔG_octanol_)/2.303RT, where ΔG values are the Gibbs energies of solvation in the relevant solvents [15]. The results are presented in Table 8.

A positive value of logP_ow_ indicates the hydrophobicity of Au(CN)_4_^−^ and while high lipophilicity is often preferred in drug development, particularly for intracellular targets, it may also hamper the dissolution of this potential active pharmaceutical ingredient. However, we have shown that gold(III) tetracyanide can form intermolecular interactions (Table 4, Table 5 and Table 7) exceeding the coordination number of Au(III) to five or even six. Therefore, when Au(CN)_4_^−^ is delivered in a form of complexes such as TG701, TG702, or TG703 it may boost its efficiency. In those compounds, the gold(III) tetracyanide forms additional coordinate bonds with Lewis bases via Se or B atoms. Additionally, the presence of amphiphilic substituted alkyl chains in those ligands is beneficial for the stability of the lipophilic complex such as Au(CN)_4_^−^. Those flexible alkyl tails of medium length can surround the gold(III) tetracyanide, serving as a drug carrier, and forming micelles. This is in agreement with the experimental results that show the superiority of TGS 701, TGS 702, and TGS 703 over TGS 512. Particularly, due to the presence of both Lewis bases and amphiphilic drug carriers in the TGS 703, both its solubility and permeability are boosted by the excipients of this formulation. The molecular modeling results suggest that in the TGS 703 form the Au(CN)_4_^−^ should be most potent, which is directly reflected in the biological analysis.

To confirm the dynamic stability and assess the dynamic behavior of Au(CN)_4_^−^, molecular dynamics (MD) simulations at the DFT level were performed. Analysis of the trajectory showed that the complex is stable at 300K as the dissociations of ligands were not observed. The structure remained square-planar. Introducing thermal motions resulted in a slight decrease in the Au-C bond length as the simulation mean was found to be 1.964 Å (Appendix A), compared to the 2.016 Å from the geometry optimization. Bonds were normally distributed (Appendix A), oscillating in the range of 1.86–2.06 Å during the entire simulation (Appendix A). Additionally, the angles remained right during the MD, with a mean of 90 degrees (Appendix A). During the simulations, the angles were oscillating in the range of 82–92 deg (Appendix A). Mean square displacement was found to be linearly correlated (R^2^ = 0.9634) with the simulation time (Appendix A), indicating that only normal diffusion occurs, characterized by the isotropic diffusion coefficient 0.0938 Å^2^ ps^−1^.

### 2.3. TGS 703 Attenuated Colitis in DSS-Induced Mouse Model

Based on in vitro and in silico studies we selected TGS 703, which was characterized by the greatest anti-inflammatory potential, to be tested in vivo. In the DSS-induced mouse model of colitis, TGS 703 was tested in two doses, 1.68 μg/kg (A) and 16.8 μg/kg (B) (both intragastrically), once daily from day 3 to day 6 after colitis induction.

DSS treatment induced colitis, as evidenced by the increased macroscopic score (9.40 ± 0.69 for DSS-only treated mice vs. 1.87 ± 0.35 for control animals, *p* < 0.001) (Figure 8A), which is consistent with our previous observations [7]. The p.o. administration of TGS 703, in both doses, significantly decreased the macroscopic score in DSS-treated animals (TGS 703 A—9.5 ± 0.69, *p* < 0.001; TGS 703 B—9.46 ± 0.42, *p* < 0.001).

Similarly, stool score was increased for DSS-only mice (0.75 ± 0.17 for control vs. 2.88 ± 0.25 for DSS-treated animals, *p* < 0.001), and this effect was partially reversed by treatment with TGS 703 A (2 ± 0.13, *p* = 0.05).

The parameters taken into consideration during microscopic evaluation included muscle thickness, cell infiltration, mucosal architecture, crypts morphology, and the presence of goblet cells. DSS treatment significantly increased the microscopic score (10.92 ± 0.81 for DSS-only treated animals vs. 3.1 ± 0.10 for control animals, *p* < 0.001). Both doses of TGS 703 caused a significant decrease in the microscopic score (TGS 703 A: 7.14 ± 0.63, *p* = 0.002 and B: 6.56 ± 0.68, *p* < 0.001).

The myeloperoxidase (MPO) activity represents neutrophil infiltration of the intestine; therefore, it is the highest in inflamed tissue [7]. In line, the MPO activity was lowest in the colon of the control animals (3.67 ± 0.46 U) and DSS-treated mice, which received TGS 703 B (6.43 ± 0.63, *p* = 0.002—compared to the DSS-only group). The MPO activity was highest in the DSS-only treated mice (11.11 ± 1.08 U) (Figure 8D).

To further present the anti-inflammatory beneficial effect of the tested compounds, we assessed the changes in body weight during the course of the experiment. A smaller loss of body weight was observed in treated groups compared to DSS-only mice (Figure 8C).

Representative photos of hematoxylin and eosin staining of colon samples are presented in Figure 9.

### 2.4. TGS 703 Influenced the Antioxidant Profile in the Mouse Colon

In DSS-only treated mice heme oxygenase-1 (HO-1) level was decreased compared to the control (*p* < 0.001), which is also consistent with our previous observations [7]. Treatment with TGS 703 A and B statistically increased the HO-1 level in DSS- treated animals (Figure 10A). Similarly, in the DSS group, the activity of catalase was decreased, while treatment with TGS 703 in both doses significantly increased catalase activity (Figure 10B).

DSS administration slightly decreased the level of glutathione (GSH) in colonic tissue; however, we have not observed any statistical difference between the control group and DSS-only treated mice. Similarly, the GSSG level was not affected by DSS administration, as compared to the control group. TGS 703 B significantly decreased the level of GSSG in colonic tissue compared to DSS-only treated mice (*p* = 0.04) (Figure 10C and Figure 10D, respectively). Glutathione peroxidase (GPX) activity was significantly increased by DSS treatment (control: 32.94 ± 6.72 vs. DSS-only: 127.74 ± 10.54; *p* < 0.001), while treatment with TGS 703 B significantly lowered the GPX activity (*p* = 0.009) (Figure 10E).

Cyclooxygenase-1 (COX-1) activity was merely affected by the DSS and DSS + TGS 703 treatment (Figure 10F). Cyclooxygenase-2 (COX-2) activity was slightly decreased in the DSS group, but TGS 703 A administration reversed this effect, and TGS 703 B significantly increased COX-2 activity compared to DSS only group (*p* = 0.03) (Figure 10G).

## 3. Discussion

Although gold had high medical significance in Chinese and Middle East medicine centuries ago, scientists have only recently “rediscovered” its anti-inflammatory properties [16]. Anti-inflammatory effects have been documented in numerous gold(I) and (III) complexes. For example, the beneficial effects of gold(I) complexes were confirmed in in vivo rat edema model and in in vitro study on THP-1 macrophages activated by lipopolysaccharide (LPS). According to Tolbatov et al., gold(I) complexes can trigger apoptosis due to the inhibition of selenium- and sulfur-containing enzymes such as thioredoxin reductase (TrxR), glutathione peroxidase, cysteine protease, or glutathione-S-transferase [17]. Of note, auranofin is the only FDA-approved example of the use of gold(I) compound [18]. Its mechanisms include inhibition of TrxR and induction of endoplasic reticulum (ER) stress, which leads to increased oxidative stress and apoptosis [19].

Similarly, gold(III) complexes involving cycloaurated phosphine sulfide complexes and N6-benzyladenine, a plant hormone, were confirmed to exhibit anti-inflammatory effects. Concurrently, Krajewska et al. investigated the potential of TGS 121, a novel gold(III) complex [7], and noted its anti-inflammatory properties in in vitro LPS-stimulated RAW264.7 macrophages and in the in vivo mouse model of colitis induced with dextran sulphate sodium (DSS).

In this study, we examined the anti-inflammatory properties of a series of new gold(III) complexes in the context of IBD. We also proved that TGS 703, the gold(III) complex with the most potent anti-inflammatory effect observed in vitro and in vivo, acts through modulation of the non-enzymatic and enzymatic antioxidant systems.

In physiological conditions, anti-oxidative defense mechanisms maintain low concentrations of reactive nitrogen and oxygen species through employment of enzymes, such as SOD, GPx, and CAT, and non-enzymatic scavengers such as glutathione. A contrary situation is observed in inflammation, where an increased release of free radicals from activated macrophages and leukocytes leads to nitrosative and oxidative stress. 

Heme oxygenase (HO) enzymes are involved in heme catabolism, but also in multiple physiological processes, including anti-inflammatory mechanisms [20]. There are two functionally active isoenzymes, the HO-1 form which is inducible and HO-2 form, the constitutive one. Noteworthy, HO-1 may show cytoprotective activity as well as cytotoxic, and the anti-inflammatory effects depend on CO level [21]. HO-1 is present in the nucleus, where it influences the transcription factor NF-E2-related factor 2 (Nrf2) and impacts anti-oxidant and metabolic defenses. Of note, in the nucleus HO-1 does not exhibit enzymatic functions [22]. Takagi et al. noticed that HO-1 acted protectively in colitis induced by DSS in the mouse model [23]. They also noticed that the inhibition of HO-1 was associated with increased levels of pro-inflammatory cytokines, tumor necrosis factor-α, and interferon-γ. Zhang et al. investigated DSS-induced colitis mice treated with hemin, an inducer of HO-1, or stannum protoporphyrin IX, an inhibitor of HO-1. Results show that in the hemin-treated group, symptoms and histologic markers of inflammation significantly ameliorated. Moreover, the number of Th17 cells decreased and the number of regulatory T cells (Treg) increased [24], which is consistent with the anti-inflammatory potential of IBD. [25,26]. In our study, mice treated with TGS 703 B were characterized by the alleviation of colitis and a significant increase in HO-1 concentration compared to the DSS-only treated mice. The observed correlation between the amelioration of colitis and an increased level of HO-1 is in line with other studies.

Catalase belongs to protective antioxidant enzymes; it acts against the devastating effects of reactive oxygen species. Catalase is responsible for the decomposition of H_2_O_2_ to O_2_ and H_2_O and is crucial in maintaining an appropriate level of H_2_O_2_ [27]. Kim et al. investigated DSS-induced colitis in mice, which were transfected with an atypical commensal *Escherichia coli* strain, which implicated extra catalase activity [28]. Mice with atypical *E. coli* displayed a significantly decreased disease score and received improved scores in histological evaluation [26]. In this study, we observed a decrease in catalase activity in DSS-only treated mice and this effect was reversed in mice treated with TGS 703 in both doses, which corroborates earlier data.

Glutathione (GSH) plays a significant role in controlling the intracellular balance of reduction-oxidation and also influences the signaling pathways affected by oxidative stress [29]. When GSH reacts with reactive oxygen species, it transforms into glutathione disulfide (GSSG); the reaction is catalyzed by GSH peroxidase (GPx). In acetic acid-induced UC in albino rats, colonic levels of GPx significantly decreased in the colitis group in comparison with the control group [30]. Vassilyadi et al. conducted a study on piglets with DSS-induced colitis, indicating a lower level of GSH and decreased synthesis of this peptide in inflammation [31]. In a rat model proposed by Wang et al., colitis was induced by a 2,4,6-trinitrobenzenesulfonic acid-ethanol solution. It was noticed that the level of GSH negatively correlates with inflammatory markers [32]. Similarly, we observed a decrease in the level of GSH in DSS-only treated mice, while treatment of inflamed mice with TGS 703 slightly increased GSH levels. Moreover, administration of TGS 703 significantly decreased the GSSG level, while a slight increase was observed in DSS-only mice. Concurrently, DSS treatment induced an increase in GPx level, and TGS 703 at the higher dose significantly reversed this effect.

The cyclooxygenase (COX) enzyme family constitutes two basic isoforms, COX-1 and COX-2. COX-1 is constitutively expressed in most human tissues and is proven to play a significant role in maintaining cell integrity in the intestinal tract. On the other hand, COX-2 is mostly induced in pathologic conditions, such as inflammatory sites and neoplasia. It was reported that COX-1 is reduced in colonic specimens obtained from IBD patients [33,34]. Correspondingly, mRNA and protein levels of COX-1 are decreased in DSS-treated mice [33]. Surprisingly, we observed a slight increase in COX-1 levels in the DSS and TGS 703 + DSS groups. Contrary, the COX-2 level was slightly decreased in the DSS group and increased after TGS 703 administration. Further investigations are needed to verify these observations.

## 4. Materials and Methods

### 4.1. Synthesis of Novel Gold(III) Complexes

TGS 404

The chlorite-cyanide complexes of monoionic gold(III) was prepared as follows: 1000 mg of 99.99% pure metallic gold was dissolved in *aqua regia* (a mixture of concentrated hydrochloric and nitric acids in a molar ratio of 3:1). Following that, gold(III) occurred in the form of very large clusters with metallic bonds (Au-Au) > 11. The water-soluble gold(III) clusters obtained were acidified with 350 cm^3^ of concentrated (36%) hydrochloric acid, and then the mixture was boiled until the volume was reduced to 20–30 cm^3^. After adding 350 cm^3^ of concentrated hydrochloric acid, the mixture was heated to the boiling point and NOCl (nitrosyl chloride) vapors were released. The above action was repeated many times until nitric acid and its oxides were completely evaporated and the gold(III) chlorides remained in the flask.

A thermostatic polyglycol bath was used to evaporate the liquids (acids) from the gold(III) salts. Obtained dry salts were re-dissolved in aqua regia. The above chemical treatment makes it possible to obtain clusters of gold(III) chloride smaller than 11 atoms.

Then, 450 cm^3^ of 6 M hydrochloric acid were added to the dry salt, and the mixture was heated again to the boiling point of the liquid and evaporated until dry salts were formed. This operation was repeated six times to obtain the smallest clusters of gold(III) until an orange-red salt of gold(III) chloride was obtained. Analysis was performed to confirm the presence of a practically pure Au_2_Cl_6_ compound. Following that, 9 g of sodium chloride (NaCl) were added to the Au_2_Cl_6_ and topped up with distilled water to approximately 500 cm^3^. Next, the mixture was boiled for several hours to obtain a compound with the formula Na_2_Au_2_Cl_8_. Then, the aqueous solution of NaCl and salt was evaporated until dry salt precipitated. Afterward, the salts were treated with 200 cm^3^ of distilled water and 600 cm^3^ of 6 M hydrochloric acid until no further color change was visible.

Following the last treatment with 6 M hydrochloric acid and its final evaporating, dry salts were obtained, which were then diluted with 800 cm^3^ of distilled water, enabling to obtain a solution of HAuCl_2_*H_2_O monatomic gold salt with a pH of approximately 1. Then, 1 M sodium hydroxide was added in order to neutralize the solution to pH 4–5. Afterward, 3.5 g of 25% dimethyl sulfoxide (SO(CH_3_)_2_) were added to obtain a water-soluble, stable complex of gold(III) with dimethyl sulfoxide with the formula AuCl_4_*SO(CH_3_)_2_*NaCl. Next, this complex was neutralized with 5% sodium bicarbonate (NaHCO_3_) to a pH of about 7.6, and then 60 g of 0.5 M water-alcoholic solution of sodium cyanide were added. The mixture was stirred at 35 °C for 2 h and then acidified with 0.2 M hydrochloric acid. The above synthesis was again stirred under reduced pressure for 4 h to drive off the free hydrogen cyanide. Such highly water-soluble monoionic gold(III) complexes were neutralized to pH 7.4 (pH of blood and lymph) with 0.1 M sodium hydroxide. The mixture was supplemented with redistilled water to a volume of 1 dm^3^ where there was 1000 mg of 5 mM monoionic gold(III).

TGS 512

TGS 512 complex is a pharmaceutical composition from two gold(III) complexes suspended in polyunsaturated fatty acid derivatives.

This composition was prepared as follows: 300 g of 0.75 M polyethylene glycol and 2 g of sodium ethoxide (alkaline catalyst) were added to 210 g of 0.25 M extra pure rapeseed oil. Next, the mixture was heated to a temperature of 92–94 °C for 3 h until a light yellow water was formed. Afterward, 2 g of 10% water solution of the above mixture was mixed with 200 cm^3^ of demineralized water. Following that, 250 cm^3^ of monoionic gold(III) made according to the patent PL232677 were added. The whole mixture was topped up to 500 cm^3^ and thoroughly mixed.

TGS 701

The emulsifying base of TGS 701 compounds is monoglycerides of epoxidized soybean oil.

MEOS Formula (Epoxidized Soybean Oil Monoglycerides):



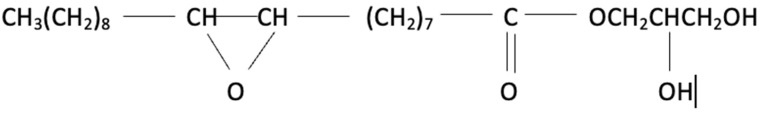



The monoglycerides of epoxidized soybean oil (MEOS) were prepared as follows: 633 g of 2/3 M epoxidized soybean oil (formula: C_57_H_106_0_10_) was mixed with 92 g 1 M glycerol and 2 g of sodium ethoxide (alkaline catalyst). The mixture was heated to 90 °C ± 1 °C for 2–3 h until a single phase of MEOS was formed with an excess of free glycerol (about 2/3 M) that could be easily separated.

Then, 15 cm^3^ of the emulsifying base of the monoglyceride phase obtained in the above reaction were mixed with 150 cm^3^ of organic silicon with boron and synthesized according to the patent PL214750 with the following formula: [CH_3_Si(ONa)_3_]_2_*C_18_H_15_BO_21_

Afterward, this homogeneous mixture was mixed with 32 cm^3^ of monoionic gold(III), made according to the Polish patent PL232677 example 1, (point a–j), containing 3.2 mg of monoionic gold(III). The whole was stabilized with a food thickener to obtain a creamy white pharmaceutical form. 

TGS 702

TGS 702 was prepared as follows: 378 g of 0.1 M 10% MEOS dissolved in 96% ethyl alcohol were added to 19.6 g 0.1 M L-selenomethionine. The whole was heated to 82–83 °C for three hours until a uniform solution of L-selenomethionine and MEOS was formed in an ethyl alcohol solution. The mixture was topped up with water to 575 g and the pH was previously neutralized to 7.5 with 1 M sodium hydroxide.

Then, 37.5 cm^3^ of monoionic gold(III), made according to the Polish patent PL232677, were poured. The solution was stirred for 15 min until a uniform emulsion was formed. The whole was stabilized with a food thickener to obtain a creamy white pharmaceutical form. 

TGS 703

TGS 703 was prepared as follows: 378 g of 0.1 M 10% MEOS dissolved in 96% ethyl alcohol were added to 129 g of 0.1 M 10% aqueous selenic acid (IV) solution. The mixture was heated to the boiling point at 90 °C for four hours until a clear, uniform solution of selenic acid (VI) and MEOS was formed in a water-alcoholic solution. After cooling, the mixture was neutralized to pH 7.0 with 1 M sodium hydroxide and supplemented with water to the weight of 507 g. Then, 250 cm^3^ of water were mixed with 32 cm^3^ of monoionic gold(III), made according to the Polish patent PL232677. The mixture was stabilized with a food thickener to obtain a white emulsion pharmaceutical form. 

### 4.2. Molecular Modeling

Density functional theory (DFT) calculations were performed using the Gaussian 16 software. All electron computations were done employing the def2-TZVP basis set and TPSS functional, with Grimme’s dispersion force corrections (TPSS-D3). This method has been recently proven to provide very accurate results for the calculations of gold compounds in a benchmark study [35].

The polarizable continuum model (PCM) was used for implicit solvation, choosing either water, n-octanol, or dimethyl sulfoxide (DMSO) as the solvent (dielectric constants equal 78.540, 9.8629, and 46.826, respectively) [36].

The natural mode frequencies were calculated in harmonic approximation to confirm that each structure was not in a transition state. The existence of only positive frequencies confirmed the nature of the stationary points on the potential surface. Additionally, the computation of vibrational frequencies allowed to estimate zero-point vibrational energy (ZPVE) corrections and thermodynamic parameters, including Gibbs free energy (ΔG) at 298.15 K and 101.325 kPa.

Molecular dynamics simulations at the DFT level were performed using the DMol3 software, choosing NVT ensemble-fixed volume with a thermostat to maintain a constant temperature. All electron relativistic computations were done employing the DNP+ basis set and TPSS functional, with Grimme’s dispersion force corrections (TPSS-D3).

The conductor-like screening model (COSMO) was used for implicit solvation, choosing water as the solvent with dielectric constant equals 78.540. Constant temperature 300 K was controlled during simulations using a massive, generalized Gaussian moments (massive GGM) thermostat with chain length 2, relaxation time 10, and Yoshida parameter 3. The time step during simulations was set to 1.0 fs, and the total simulation time was set to 10 ps.

### 4.3. Cell Line Culture

RAW264.7 murine macrophage cells (ATCC TIB-71) were cultured in a humidified atmosphere of 5% CO_2_ at 37 °C. Cells were cultured in Dulbecco’s modified eagle medium (Gibco, Waltham, MA, USA) supplemented with 10% bovine calf serum, 25 mM HEPES, 0.5% penicillin-streptomycin, 2 mM Ala-Gln, and 1 mM sodium pyruvate. The culture medium was replaced every 3 days, and the cells were trypsinized at approximately 80% confluence.

### 4.4. Cytotoxicity Assessment

Neutral red uptake (NRU) assay was performed to analyze the cytotoxicity of analyzed complexes on RAW264.7 cell line cultures after 48 h of exposure to the studied compounds. Complexes were tested in the range of concentrations 1 × 10^−8^ to 1 × 10^−3^.

The experiments were performed according to the protocol presented previously [7]. Briefly, the cells were seeded on 96-well plates (20,000 per well) and incubated for 48 h with or without analyzed compounds. After incubation, the medium was discarded and 100 μL per well of 0.05 mg/mL NR solution in culture medium was added. After 1 h incubation, the cells were washed once with PBS (pH 7.4), and 100 μL per well of dye solvent (40% ethanol and 10% acetic acid in water) was added. The plates were shaken for 10 min on a rotary shaker, and the absorbance was measured at 540 nm in a microplate reader (iMARK Microplate Reader, Biorad, Hertfordshire, UK) using blank as a reference. Cytotoxicity was expressed as a percentage of negative control (medium without the studied complex).

### 4.5. Griess Assay

Griess assay was performed to assess the nitrite concentration in cell culture supernatants, as previously described [7]. The cells were seeded on 96-well plates (20,000 per well) and incubated for 24 h with medium only (control) or 1 μg/mL LPS with or without studied compounds. After 24 h, the medium was discarded and solutions containing TGS derivatives were added. After another 24 h, 40 mg/mL Griess reagent water solution was mixed with an equal volume of cell culture supernatant and incubated for 15 min in the dark; the absorbance was read at 540 nm.

### 4.6. Animals

Male BALBc mice (22–28 g; approximately 8 weeks of age) were obtained from the Animal Facility of University of Lodz, Poland. The animals were housed at 22–23 °C and maintained under a 12 h light/dark cycle with constant access to laboratory chow and tap water. Animal protocols were approved by the Local Ethical Committee for Animal Research at the Medical University of Lodz (#4/ŁB85/2018 and #13/ŁB130/2019). All efforts were made to minimize animal suffering and to reduce the number of animals used.

### 4.7. Induction of Colitis

Colitis was induced by administration of dextran sulphate sodium (DSS; molecular weight 40,000, Biochemica, PanReac AppliChem, Darmstadt, Germany), as described earlier [4]. Mice (n = 7–9 per experimental group) received DSS (4% wt/vol) in drinking water from day 0 to day 5. On days 6 and 7 the animals received water. Control mice were given drinking water throughout the whole experiment. Animal body weight was evaluated daily. Mice were euthanized by cervical dislocation on day 7, and colonic damage was assessed.

### 4.8. Pharmacological Treatment

TGS 703 was administered intragastrically once daily from day 3 to day 6 of the experiment, 100 μL per mouse. The doses were corresponding to our earlier study of TGS 121 [4]. Similarly, we used TGS 703 at dose A—1.68 μg/kg and dose B—16.8 μg/kg. Control and DSS groups received 0.9% NaCl alone (100 μL) administered intragastrically.

### 4.9. Microscopic Score and Colonic Damage Evaluation

The colons were weighed, dissected longitudinally, and the feces were removed. The well-established semiquantitative system was used for the macroscopic damage scoring, as described earlier [4]. Distal colon fragments (approximately 0.5 cm in length) were then fixed in 10% neutral-buffered formalin for at least 24 h at 4 °C, dehydrated in sucrose, embedded in paraffin, sectioned at 5 μm, and mounted onto slides. The hematoxylin-eosin stained sections were scored using a Zeiss Axio Imager setup (Jena, Germany), according to the microscopic total damage score system described earlier [4].

### 4.10. Determination of Tissue Myeloperoxidase Activity

Myeloperoxidase (MPO) activity was determined as described earlier [19]. Briefly, colon fragments (approx. 30 mg) were homogenized in hexadecyltrimethylammonium bromide (HTAB) buffer (0.5% HTAB in 50 mM potassium phosphate buffer, pH 6.0; 50 mg tissue/mL) and centrifuged (15 min, 13,200× *g*, 4 °C). On a 96-well plate, 200 μL of 50 mM potassium phosphate buffer (pH 6.0), supplemented with 0.167 mg/mL of O-dianisidine hydrochloride and 0.05 μL of 1% hydrogen peroxide, was added to 7 μL of the supernatant. Absorbance was measured at 450 nm at 30 and 60 s (iMARK Microplate Reader, Biorad, Hertfordshire, UK). MPO activity was expressed in milliunits per gram of wet tissue; 1 unit being the quantity of enzyme able to convert 1 μmol hydrogen peroxide to water in 1 min at room temperature.

### 4.11. Statistical Analysis

Results were analyzed in Prism 9 (GraphPad Software Inc., La Jolla, CA, USA) with the use of one-way ANOVA followed by the Tukey post hoc test. The data are presented as mean ± standard error of the mean (SEM). *p* values < 0.05 were considered statistically significant.

## 5. Conclusions

In conclusion, in this study, we report on a new class of gold(III) derivatives with anti-inflammatory potential. Of all compounds tested, TGS 703 significantly alleviated inflammation in the in vivo model of colitis. The experimental findings were also supported, at the molecular level, by the results of molecular modeling calculations. Based on the presented results, we state that the tested compounds are an attractive therapeutic option for inflammation and IBD, and further pre-clinical investigations should be performed. Future studies on specific mechanisms and novel derivatives may give a strong rationale for possible clinical translation.

## Figures and Tables

**Figure 1 ijms-24-07025-f001:**
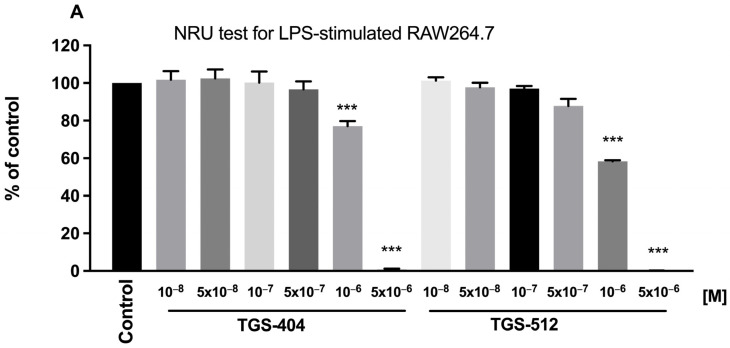
Neutral red uptake test (**A**,**B**) and Griess test (**C**,**D**) for RAW264.7 macrophages treated with TGS compounds. ## *p* < 0.01, ### *p* < 0.001 vs. control; *** *p* < 0.001 vs. 1 µg/mL lipopolysaccharide (LPS).

**Figure 2 ijms-24-07025-f002:**
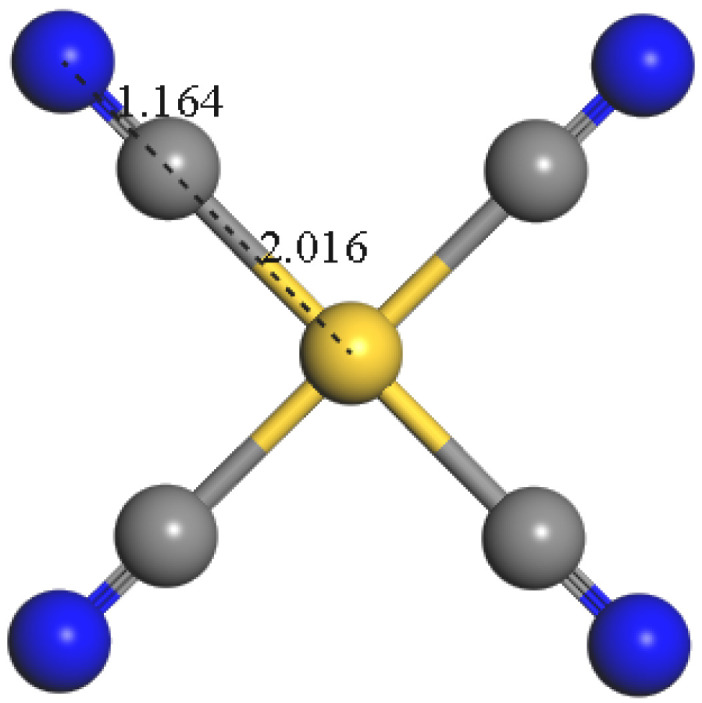
The structure of the most stable gold(III) tetracyanide isomer, Au(CN)4^−^. The values indicate the N-C (1.164 Å) and C-Au (2.106 Å) lengths. Atom coloring: Au—gold, C—grey, N—blue.

**Figure 3 ijms-24-07025-f003:**
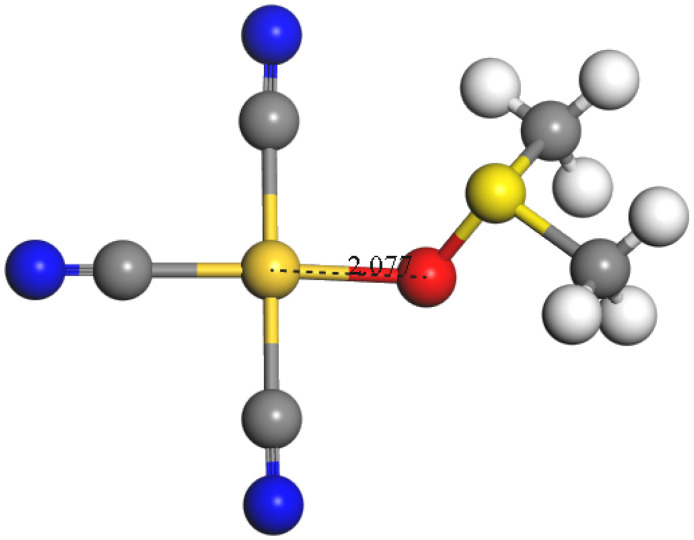
The structure of the most stable gold(III) tricyanide dimethyl sulfide complex isomer, [Au(CN)_3_DMSO] κ-O. The value indicates the Au-O (2.077 Å) length. Atom coloring: Au—gold, C—grey, N—blue, O—red, S—yellow, and H—white. It should be noted that the substitution of one CN^−^ ligand by DMSO is both energetically and thermodynamically not favorable.

**Figure 4 ijms-24-07025-f004:**
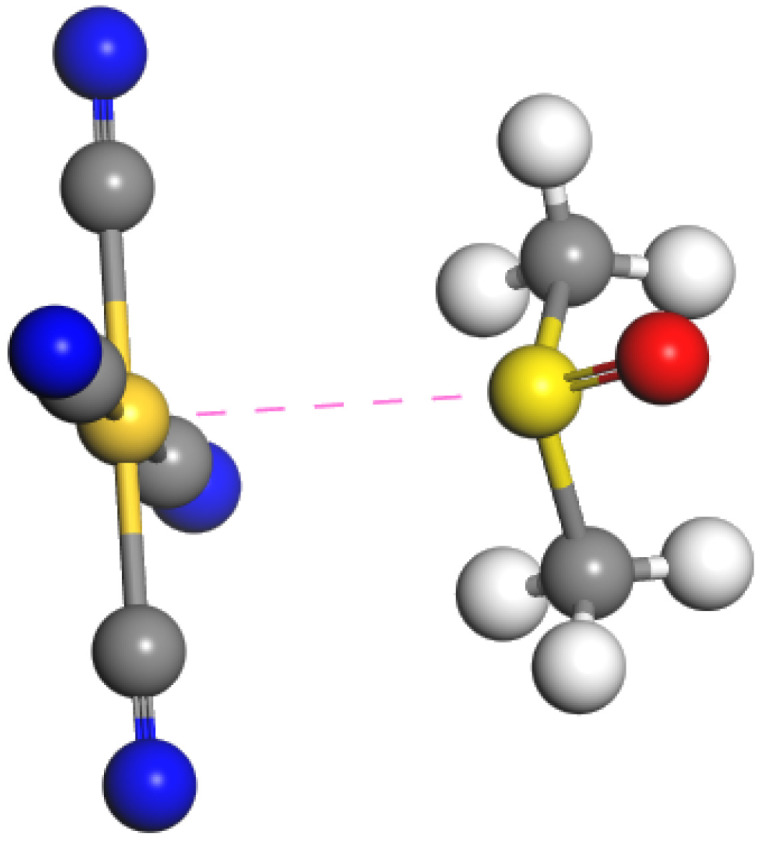
The structure of the most stable gold(III) tetracyanide dimethyl sulfide complex isomer, [Au(CN)_4_DMSO]^−^ κ-S. Atom coloring: Au—gold, C—grey, N—blue, S—yellow, O—red, and H—white. The pink dashed line represents the close contact between Au and S atoms. It should be noted that while extending the coordination number of Au(III) above four, the sulfur atom forms an energetically more favorable interaction with the central ion, instead of the oxygen. This is in opposition to the complex when the CN^−^ ion was substituted by DMSO, as presented in Figure 3.

**Figure 5 ijms-24-07025-f005:**
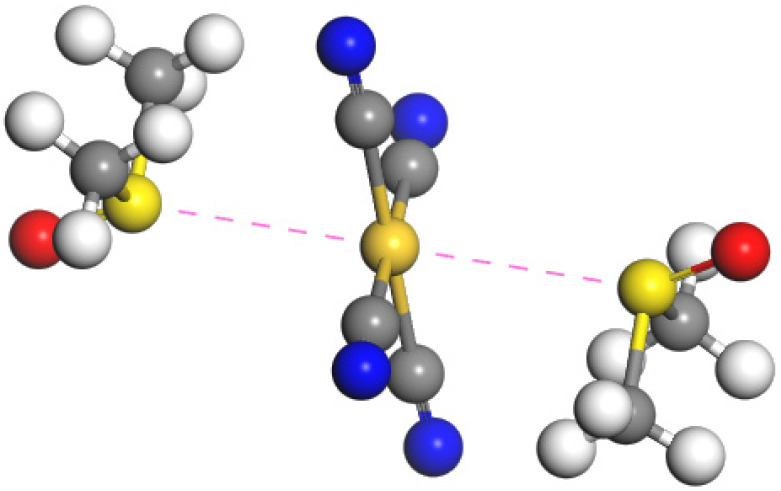
The structure of the most stable gold(III) tetracyanide: dimethyl sulfide (1:2) complex isomer, [Au(CN)_4_DMSO_2_]^−^ κ-S, S. Atom coloring: Au—gold, C—grey, N—blue, S—yellow, O—red, and H—white. The pink dashed line represents close contact between Au and S atoms. Although the stabilizing interactions between the additional ligands, DMSO molecules, and gold(III) tetracyanide are being formed, the presented complex is not stable due to the entropic penalty resulting in the positive value of free enthalpy of formation. The results of the calculations indicate the preference of the κ-S, S isomer. However, unexpectedly, the κ-S, O isomer was found to be the least stable one. Similarly, as in the case of mono DMSO complexes, the formation of the studied molecules is energetically favorable due to the formation of the intermolecular interactions between the positively charged Au(III) and DMSO, which is the Lewis base. However, due to the decrease of entropy resulting from such complexation, the formation of both mono and di DMSO complexes is not preferable thermodynamically due to the positive ΔG of such reactions.

**Figure 6 ijms-24-07025-f006:**
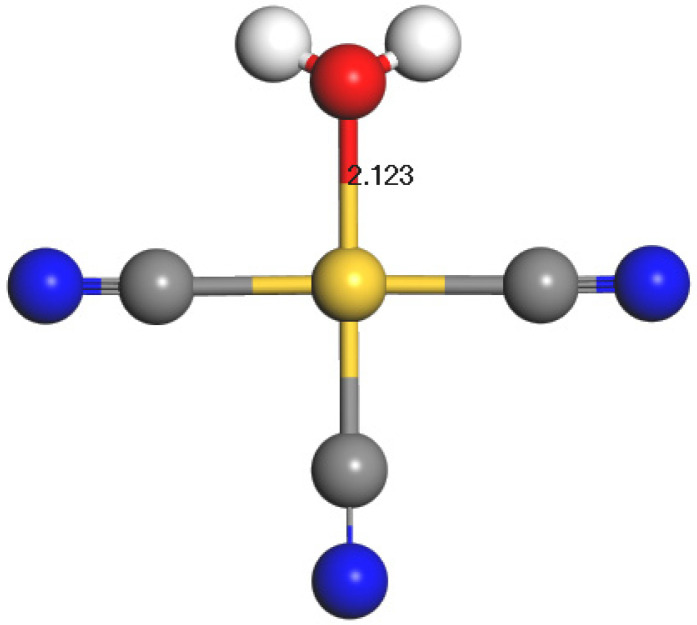
The structure of the gold(III) tricyanide aquo complex, [Au(CN)_3_H_2_O]. The value indicates the Au-O (2.123 Å) length. Atom coloring: Au—gold, C—grey, O—red, N—blue, and H—white. It should be noted that although the complex with water serving as ligand is stable, the substitution of CN^−^ ligand by water is both energetically and thermodynamically not favorable. This explains why the dissociation of gold(III) tetracyanide in water environment is not observed.

**Figure 7 ijms-24-07025-f007:**
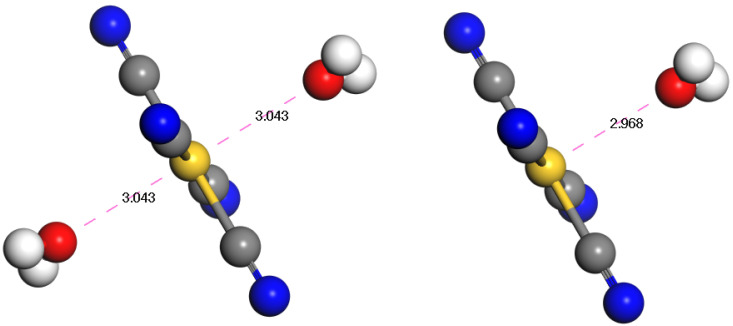
The structures of the gold(III) tetracyanide aquo (**right**) and diaquo (**left**) complexes, [Au(CN)_4_H_2_O]^−^ and [Au(CN)_4_(H_2_O)_2_]^−^. Atom coloring: Au—gold, C—grey, N—blue, O—red, and H—white. Despite the intermolecular electrostatic interactions occurring between the positively charged Au atom of Au(CN)_4_^−^ and oxygen atoms of water, the decrease of entropy destabilizes those aquo complexes.

**Figure 8 ijms-24-07025-f008:**
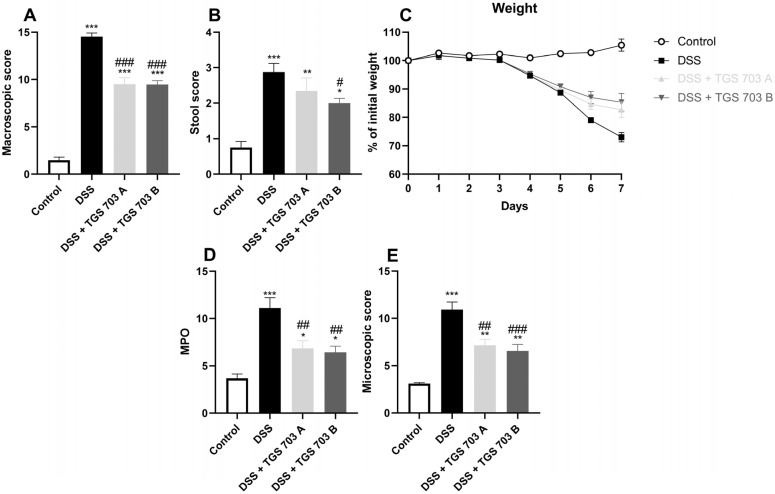
The effect of TGS 703 on colonic inflammation in a dextran sulphate sodium (DSS)-induced model of colitis: macroscopic score (**A**), stool score (**B**), myeloperoxidase activity (**D**), and microscopic score (**E**) for control, DSS-only treated mice, and mice with DSS-induced colitis treated with TGS 703 in two doses: 1.68 μg/kg (TGS 703 0.1A) and 16.8 μg/kg (TGS 703 1A). * *p* < 0.05, ** *p* < 0.01 and *** *p* < 0.001 as compared with the control mice; # *p* < 0.05, ## *p* < 0.01, ### *p* < 0.001 as compared with DSS-treated animals. (**C**) Presents changes in the weight of mice during the DSS-induced animal model of colitis.

**Figure 9 ijms-24-07025-f009:**
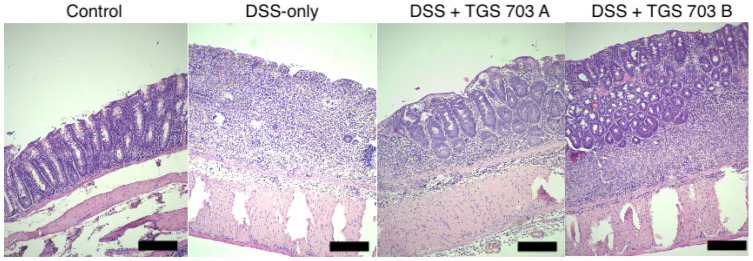
Representative photos of hematoxylin and eosin staining of colon samples. Scale bar = 100 μm.

**Figure 10 ijms-24-07025-f010:**
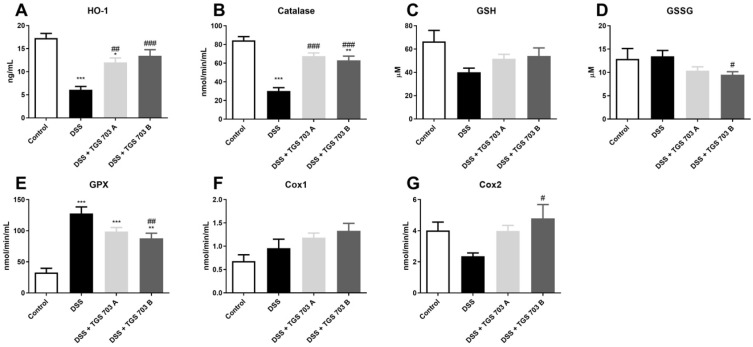
The influence of DSS colitis and TGS 703 treatment on the antioxidant profile in the mouse colon: heme oxygenase-1 (**A**), catalase (**B**), glutathione (**C**), glutathione disulfide (**D**), glutathione peroxidase (**E**), cyclooxygenase-1 (**F**), and cyclooxygenase-2 (**G**) for control, DSS-only treated mice, and mice with DSS-induced colitis treated with TGS 703 in two doses: 1.68 μg/kg (TGS 701 0.1A) and 16.8 μg/kg (TGS 703 1A). * *p* < 0.05, ** *p* < 0.01, *** *p* < 0.001 as compared with the control mice; # *p* < 0.05, ## *p* < 0.01, ### *p* < 0.001 as compared with DSS-treated animals.

**Table 1 ijms-24-07025-t001:** Energy (E); energy corrected by the zero-point vibrational energy (E + ZPVE); Gibbs free energy (ΔG) of the complex formation of gold(III) tetracyanide. In vacuo—results obtained without solvation. Water and DMSO—results obtained using PCM implicit solvation model. All values are in kcal/mol.

		Au(CN)_4_^−^
**E**	In vacuo	−1539.48
	Water	−493.08
	DMSO	−502.14
**E + ZPVE**	In vacuo	−1531.36
	Water	−485.17
	DMSO	−494.21
**ΔG**	In vacuo	−1495.58
	Water	−449.70
	DMSO	−458.74

**Table 2 ijms-24-07025-t002:** Energy (E); energy corrected by the zero-point vibrational energy (E + ZPVE); Gibbs free energy (ΔG) of the linkage isomers of gold(III) tetracyanide. In vacuo—results obtained without solvation. Water and DMSO—results obtained using PCM implicit solvation model. NS indicates the non-symmetric isomer in which the CN^−^ are next to each other. S indicates the symmetric isomer in which the CN^−^ are opposite to each other. All values are in kcal/mol. The values in each row are provided relative to the Au(CN)_4_^−^.

		Au(CN)_4_^−^	Au(CN)_3_(NC)^−^	Au(CN)_2_(NC)_2_^−^ NS	Au(CN)_2_(NC)_2_^−^ S	Au(NC)_3_CN^−^	Au(NC)_4_^−^
**E**	In vacuo	0.00	17.05	33.36	37.08	52.65	71.26
	Water	0.00	17.25	33.77	37.71	53.48	72.50
	DMSO	0.00	17.25	33.77	37.68	53.47	72.48
**E +** **ZPVE**	In vacuo	0.00	16.69	32.64	36.33	51.55	69.73
	Water	0.00	17.06	33.19	37.14	52.58	71.13
	DMSO	0.00	17.06	33.18	37.13	52.57	71.13
**ΔG**	In vacuo	0.00	16.56	32.39	36.07	51.17	69.16
	Water	0.00	17.20	33.11	37.17	52.52	70.81
	DMSO	0.00	17.20	33.10	37.21	52.51	70.84

**Table 3 ijms-24-07025-t003:** Energy (E); energy corrected by the zero-point vibrational energy (E + ZPVE); Gibbs free energy (ΔG) of the complex formation of gold(III) tricyanide dimethyl sulfide complex. In vacuo—results obtained without solvation. DMSO—results obtained using PCM implicit solvation model. All values are in kcal/mol.

	[Au(CN)_3_DMSO] κ-S	[Au(CN)_3_DMSO] κ-O	[Au(CN)_3_DMSO] κ-S	[Au(CN)_3_DMSO] κ-O
	DMSO		In Vacuo	
**E**	34.42	34.20	73.20	72.77
**E + ZPVE**	34.32	33.71	118.19	72.07
**ΔG**	37.04	35.69	120.51	73.82

**Table 4 ijms-24-07025-t004:** Energy (E); energy corrected by the zero-point vibrational energy (E + ZPVE); Gibbs free energy (ΔG) of the complex formation of gold(III) tetracyanide dimethyl sulfide (1:1) complex. In vacuo—results obtained without solvation. DMSO—results obtained using PCM implicit solvation model. All values are in kcal/mol.

	[Au(CN)_4_DMSO]^−^ κ-S	[Au(CN)_4_DMSO]^−^ κ-O	[Au(CN)_4_DMSO]^−^ κ-S	[Au(CN)_4_DMSO]^−^ κ-O
	DMSO		In Vacuo	
**E**	−4.96	−5.24	−10.15	−4.89
**E + ZPVE**	−4.39	−4.73	−9.53	−4.43
**ΔG**	4.26	6.20	0.03	5.54

**Table 5 ijms-24-07025-t005:** Energy (E); energy corrected by the zero-point vibrational energy (E + ZPVE); Gibbs free energy (ΔG) of the complex formation of gold(III) tetracyanide dimethyl sulfide (1:2) complex. In vacuo—results obtained without solvation. DMSO—results obtained using PCM implicit solvation model. All values are in kcal/mol.

	[Au(CN)_4_DMSO_2_]^−^ κ-S,S	[Au(CN)_4_DMSO_2_]^−^ κ-S,O	[Au(CN)_4_DMSO_2_]^−^ κ-O,O	[Au(CN)_4_DMSO_2_]^−^ κ-S,S	[Au(CN)_4_DMSO_2_]^−^ κ-S,O	[Au(CN)_4_DMSO_2_]^−^ κ-O,O
	DMSO			In Vacuo		
**E**	−10.07	−0.08	−10.01	−20.56	−14.85	−15.45
**E + ZPVE**	−9.00	0.03	−8.89	−19.28	−14.02	−14.39
**ΔG**	9.81	19.33	10.54	−0.05	7.29	5.03

**Table 6 ijms-24-07025-t006:** Energy (E); energy corrected by the zero-point vibrational energy (E + ZPVE); Gibbs free energy (ΔG) of the complex formation of gold(III) tricyanide aquo complex. In vacuo—results obtained without solvation. Water—results obtained using PCM implicit solvation model. All values are in kcal/mol.

	[Au(CN)_3_H_2_O]	[Au(CN)_3_H_2_O]
	Water	In Vacuo
**E**	46.11	87.02
**E + ZPVE**	47.19	87.67
**ΔG**	46.70	86.96

**Table 7 ijms-24-07025-t007:** Energy (E); energy corrected by the zero-point vibrational energy (E + ZPVE); Gibbs free energy (ΔG) of the complex formation of gold(III) tetracyanide aquo and diaquo complexes. In vacuo—results obtained without solvation. Water—results obtained using PCM implicit solvation model. All values are in kcal/mol.

	[Au(CN)_4_H_2_O]^−^	[Au(CN)_4_(H_2_O)_2_]^−^	[Au(CN)_4_H_2_O]^−^	[Au(CN)_4_(H_2_O)_2_]^−^
	Water		In Vacuo	
**E**	−2.06	−7.55	−1.36	−1.75
**E + ZPVE**	−1.37	−5.20	−0.90	−1.08
**ΔG**	5.80	11.07	7.33	13.44

**Table 8 ijms-24-07025-t008:** ΔG_water_, ΔG_octanol_ the Gibbs energies of solvation in water and octanol [kcal/mol]. logP_OW_—the octanol/water partition coefficient.

ΔG_water_	ΔG_octanol_	logP_OW_
−46.12	−41.40	3.46

## Data Availability

Data is unavailable due to privacy restrictions.

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
