# Peer review of "A New Gold(III) Complex, TGS 703, Shows Potent Anti-Inflammatory Activity in Colitis via the Enzymatic and Non-Enzymatic Antioxidant System—An In Vitro, In Silico, and In Vivo Study"

_ijms, 2023, doi:10.3390/ijms24087025_

Round 1

Reviewer 1 Report

In this work, Vlodarczyk et al study the anti-inflammatory effects of new gold (III) complexes by using in vitro, in silico and in vivo studies. Several new gold (III) complexes (TGS 404, 512, 701, 702, 703) were designed and tested in Lipopolysaccharide (LPS)-stimulated RAW264.7 cells to study cytotoxicity and NO production. In silico modeling was used to study the gold complexes structure vs. their activity and stability. Dextran sulphate sodium (DSS)-induced mouse model of colitis was employed to characterize the anti-inflammatory activity in vivo. Selected based on in vitro and in silico analyses, TGS 703 significantly alleviated inflammation in the DSS-induced mouse model of colitis.

The work is of great scientific interest since it is demonstrated the anti-inflammatory effect of a new gold (III) complex TGS703 in a mouse model of colitis. The results are well presented and support the conclusions. However, some aspects should be improved or clarified.

Major comments

-The authors tested the cytotoxicity of the gold complexes in macrophages activated with LPS in a similar way to what occurs in a situation of acute inflammation. However, compound TGS 703 when administered orally in the mouse model of DSS will primarily contact intestinal epithelial cells, not macrophages. Therefore, the authors should test the cytotoxicity of the compounds in intestinal epithelial cells (Caco-2, HT-29) and not in macrophages.

-The authors test the pure gold products in in vitro assays at concentrations between 10-8-10-3 M. However, in the material and methods they describe the preparation of pharmaceutical formulations with these gold products at a different concentration. Please clarify what these formulas are made for, if the compounds are then tested at a different concentration.

-In cytotoxicity assays, the ED50 should be calculated for each compound and compared with a known compound such as cisplatin or auranofin.

-In the discussion, the authors should relate the structure and/or chemical properties of the compounds obtained in the in silico studies with the biological effects.

-In mice with inflammation, how is it explained that the catalase or COX-2 are reduced?

Minor comments

- In section 2.3. the authors should clarify that the reversal of the effects obtained with the TGS 703 is partial, since the data show that the changes are small.

-In section 4.4. On what day post-seeding are the experiments done in cells? Are the cells in confluence when you perform the experiments? When do you get 80% confluence?

-In section 4.3. the abbreviation of BCS or P/S should not be used, write the full name please.

Reviewer 2 Report

The manuscript "A new gold(III) complex TGS 703 shows potent anti-inflammatory activity in colitis via the enzymatic and non-enzymatic antioxidant system – an in vitro, in silico, and in vivo study" addresses an important issue: the search for treatments against Inflammatory bowel diseases (IBD) which constitute worldwide health-care issues. The authors performed a thorough computational study combined with an in vitro and in vivo investigation using a series of new gold (III) complexes that present anti-inflammatory potential and may have therapeutic application in the treatment of IBD.

The objectives were clearly stated and explained in the manuscript, however the experimental strategy raises some major concerns and so the experimental information from which the conclusions were drawn. The manuscript is overall well written and has good organization with minor English language and style spell check required. The authors have done a great job on analyzing the experimental data and on discussing the results and their limitations, considering always different alternative explanations/considerations for interpreting the results.

The paper is interesting but there is a need for more experimental detail in order to critically review the data. Specifically, they should provide information for the following questions and comments:

Major points:

1.      The authors should include more recent update on this topic and compare how this study further advances the current knowledge in the “Introduction section”.

2.      What is the advantage of the techniques used compared to other techniques currently used in molecular modeling?

3.      A validation using Molecular Dynamics simulations is highly encouraged.

4.      Unify the style of the references in the References Section and add DOI in the cases it is possible. And use the same reference and citation (follow MDPI’s guidelines) style in the main text.

5.      The Methods section in the study should be more accurately described for each technique used with emphasis on the in vitro and in vivo experiments which should be described in more details in the Materials & Methods Section.

6.      Have the authors checked the size of the gold nanoparticles used and their polydispersity? If yes by means of which techniques and what are the results?

Minor points:

1.      Captions in several Figures are scarce, a more detailed description is needed specially for Figures 3-7.

2.      The terms in vitro, in silico and in vivo should be italicized every time they are used.

3.      The resolution and quality of some Figures is low, the authors should provide higher quality Figures specially for Figure 8F.

Round 2

Reviewer 1 Report

The authors have clarified all the points and improved the manuscript.